# The Impact of Global Budgeting on the Efficiency of Healthcare under a Single-Payer System in Taiwan

**DOI:** 10.3390/ijerph182010983

**Published:** 2021-10-19

**Authors:** Shao-Wei Yang, Kuo-Chung Chu, Victor B. Kreng

**Affiliations:** 1Department of Information Management, National Taipei University of Nursing and Health Sciences, Taipei City 112, Taiwan; shaowei@ntunhs.edu.tw; 2Department of Industrial and Information Management, National Cheng Kung University, Tainan City 701, Taiwan

**Keywords:** national health insurance, data envelopment analysis, single payer, global budgeting, productivity, financial efficiency

## Abstract

Since 1995, a national health insurance (NHI) program has been in operation in Taiwan, which provides uniform comprehensive coverage. Forced by severe financial deficit, global budgeting reimbursement was adopted in the healthcare sector to control healthcare expenditures in 2002. A two-stage data envelopment analysis (DEA) approach was used to measure the efficiency of hospital resource allocation among stakeholders in Taiwan’s NHI system, and to further explore the changes in resource allocation after the introduction of a global budgeting payment scheme. The dataset was collected from the annual statistical reports of Taiwan’s Ministry of Health and Welfare (MOHW) and was used to estimate the efficiency of resource allocation in hospital-based healthcare services under global budgeting. In terms of efficiency during the period from 2003 to 2009, one-third of decision-making units (DMUs) improved their productivity in stage I, and seven out of the total of eighteen DMUs saw falls in financial efficiency in stage II. After global budgeting was implemented, there were significant positive impacts on the efficiency of hospital resource allocation in Taiwan. The two-stage DEA model for analyzing the effects of the global budgeting reimbursement system on productivity and financial efficiency represents a key decision-making tool for hospital administrators and policymakers.

## 1. Introduction

Since 1995, a national health insurance (NHI) program has been in operation in Taiwan, through which universal healthcare within a single-payer closed system is provided for more than 99% of the country’s 23 million residents. This is the case for most OECD countries which have developed healthcare systems and can offer their citizens some protection against the financial risks associated with illness. The healthcare delivery system in Taiwan is market driven with a mix of public and private owned hospitals, clinics and independent ancillary services from pharmacies and midwifery clinics, in which there are three related stakeholders: the Bureau of National Health Insurance (BNHI), health care providers and the insured.

In the early days of this system, the BNHI paid health care providers on a classic fee-for-service (FFS) basis, which had uniform national fee schedules. Accordingly, hospitals tended to increase the volume of their services to achieve better revenues, and health care providers may have also offered more extensive and expensive treatments than were needed [1]. Due to aging populations and the increased cost of healthcare services, the rising demand for health care has generated attention surrounding cost management [1,2,3].

In the face of financial pressure, global budgeting reimbursement had gained increasing popularity, and appears to be effective in arresting the untamed growth in healthcare expenditures in many OECD, North American and Asian countries [4,5,6,7,8,9]. With experiences of using global budgeting in private health insurance (PHI), the administrative costs have been shown to be 17% of all healthcare expenditures in Canada, as compared to 31% in the U.S. [10]. In contrast, the single-payer system in Canada has a stronger position than other health care providers and tends to have lower administrative costs than multiple-payer systems in the U.S. [11].

Additionally, the global budgeting reimbursement system was introduced to contain overall costs through a fixed maximum expenditure for a defined set of healthcare services while Taiwan’s BNHI faced severe financial pressure. Drawing on the experiences of Canada and Germany, the BNHI imposed global budgeting first on dental care in 1998, with an expenditure cap based on prospective price setting with a floating value mechanism; then on traditional Chinese medicine in 2000; primary care in 2001; and finally, on hospitals in mid-2002, to complete the phasing in of this approach for the entire Taiwan healthcare system [12,13]. Furthermore, healthcare resources are allocated among providers based on capacity and historical patterns of consumption under the global budgeting system.

According to the design of global budgeting reimbursement, the budget process takes the form of a negotiation between the funding source and the healthcare sector in Taiwan’s NHI system. The budget, determined by a combination of historical expenditure levels and risk adjustments, was divided and allocated to six regional divisions across Taiwan. To prevent the continued and unsustainable growth in medical resource utilization, hospitals have been paid in relative value unit (RVU) to keep the total amount claimed under a pre-set ceiling, which means that treating more patients might not necessarily mean more revenue. Furthermore, global budgeting tends to have more control over inputs than outputs, and must thus reduce resource inputs to reach the efficiency frontier.

In addition, the payment unit for reimbursement was changed from the original “TWD” to RVU to prevent the continued and unsustainable growth in medical resource utilization. The floating point-value mechanism provides health care providers with an incentive to control the volume of service units supplied so that reimbursements do not exceed the budget that has been set. If the number of service units supplied exceeds the amount expected when the budget was determined, the point value will be less than one. Although the global budgeting reimbursement system with a floating point-value mechanism might resemble a classic zero-sum game, hospitals would rather choose actions that maximize their utility without considering the choices of others. Once the number of healthcare services exceeds the pre-set ceiling, the point value decreases. For example, the outpatient point value significantly decreased between 2002 and 2004, and the point value fell to the low point of 0.75 in 2004 (as shown in Figure 1). Wishing to contain costs, health care providers may avoid unnecessary treatments to increase the point value over the long term. Based on the financial incentive, the global budgeting payment may have a positive impact on healthcare resource allocation under Taiwan’s NHI system.

In order to investigate how the global budgeting reimbursement system influences healthcare resource allocation in Taiwan’s NHI system, it is essential to estimate the economic efficiency of production and finance. Since the explicit goal of NHI in Taiwan is to ensure equal access to healthcare, the resource allocation process focuses not only on customized health care for individuals with different requirements, but also on efficiency. The efficiency criteria thus used provide another way to assess the performance of healthcare policy with regard to achieving the goal of moving the overall healthcare system towards cost containment. In addition, the distribution of healthcare resources under the NHI system should aim to prevent significant differences between urban and rural areas. At the macro level, the efficiency criteria focus on resource allocation efficiency, which combines various inputs to produce optimal health care—in terms of quality and quantity—by optimizing the use of available resources. Accordingly, this study adopted a two-stage data envelopment analysis (DEA) approach to measure the efficiency of hospital resource allocation among stakeholders and further explore the changes in resource allocation after global budgeting was adopted by Taiwan’s NHI program to become a single-payer health insurance system.

## 2. Materials and Methods

### 2.1. Data Envelopment Analysis

The DEA method was first proposed by Charnes, Cooper and Rhodes (1978), who employed a mathematical programming model (CCR model) to measure the technical efficiency of DMUs (decision-making units) based on the Pareto optimum concept [14,15,16]. Banker, Charnes and Cooper (1984) used the four postulates of production possibility aggregation and Shephard’s distance function to construct a BCC model to measure both technical efficiency (TE) and scale efficiency (SE) [14]. A key advantage of the production correspondence implicit in the DEA is the substitution between inputs and outputs, which is important in the healthcare sector, as many hospitals often do this. For example, registered nurses can share responsibility for inoculations with physicians in the U.S. Cheng et al. (2015), Zheng et al. (2018) and Chen et al. (2020) also adopted DEA to evaluate efficiency in healthcare [17,18,19]. In addition, Kontodimpouls and Kiakas (2006) used DEA to evaluate the total factor productivity of dialysis facilities in Greece using 12-year (1993–2004) panel data to decompose the productivity into technical efficiency change, scale efficiency change and technological change [20]. Furthermore, the two-stage DEA model provides greater managerial insight into the locations of inefficiency within an organization, enabling managers to focus more attention on relatively inefficient sub-DMUs [21].

Consider a set of *n* observations of the DMUs, among which a DMU uses m inputs to generate s outputs. In Equation (1), *Yrj* is denoted as the *r*-th output of the *j*-th DMU and *Xij* as the *i*-th input of *j*-th DMU. The efficiency of *j*-th DMU, *Hj*, is a solution to the following:(1)$$\mathrm{Max}\;H_{j}=\sum_{r=1}^{s}u_{r}Y_{rj}$$

Subject to $\sum_{r=1}^{s}u_{r}Y_{rj}-\sum_{i=1}^{m}v_{i}X_{ij}\leq 0$:$$\begin{array}{c} \overset{m}{\underset{i=1}{\sum }}v_{i}X_{ij}=1\\ u_{r}≧\varepsilon ≧0\;,\;v_{i}≧\varepsilon ≧0 \end{array}$$
where *i* = *1,2,…,m*, *r = 1,2,…,s*, *j = 1,2,…,n*, and *u_r_* and *v_i_* are the given weights associated with each output and input. *H_j_* is the relative efficiency of DMU *J*, where *H_j_* = 1 indicates an efficient DMU and *H_j_* < 1 indicates an inefficient DMU.

Banker, Charnes and Cooper (1984) used the four postulates of production possibility aggregation and Shephard’s distance function to construct a BCC model to measure technical efficiency (TE) and scale efficiency (SE). Since not every industry presents a constant return to scale, an input-oriented BCC model under variable return to scale with a free variable δ is introduced in this work to minimize the inputs with the given outputs in Equation (2):(2)$$\mathrm{Max}\;H_{j}=\sum_{r=1}^{s}u_{r}Y_{rj}-\delta$$

Subject to $\sum_{i=1}^{m}v_{i}X_{ij}=1$:$$\begin{array}{c} \overset{s}{\underset{r=1}{\sum }}u_{r}Y_{rj}-\overset{m}{\underset{i=1}{\sum }}v_{i}X_{ij}-\delta \leq 0\\ u_{r}≧\varepsilon ≧0\;,\;v_{i}≧\varepsilon ≧0\\ i=\;1,2\ldots m;\;r=\;1,2\ldots s;\;j=\;1,2\ldots n, \end{array}$$
where δ, which can be either positive or negative, is the intercept of the efficient frontier on the *X* coordinate which allows the efficient frontier not to pass through the origin:(1)δ < 0: increasing returns to scale in the DMU;(2)δ = 0: constant returns to scale in the DMU;(3)δ > 0: decreasing returns to scale in the DMU.

Recently, extensions to the DEA have introduced intermediate production into the traditional model, in which the production process can be divided into two sequential processes with the outputs of the first stage serving as the inputs for the second, as shown in Figure 2. The two-stage DEA, which was proposed by Seiford and Zhu (1999), was first used to separate overall efficiency into profitability and marketability to measure the relative managerial efficiency of the top 55 U.S. commercial banks in 1995 [22]. In addition, Chilingerian and Sherman (2004) described another two-stage DEA process for measuring physician care [23].

As the producers in the healthcare triangle, health care providers first supply the patient with medical services, and then make a claim for reimbursement via the single-payer system (BNHI). The operational capability of hospital facilities and equipment is measured by overall efficiency, and efficiency in the sub-processes model is divided into the productivity of the healthcare input and the financial efficiency of the medical services. Productivity means the minimum input quantity needed to produce a fixed volume of medical services, while financial efficiency reflects that the hospitals responded to provide appropriate services to patients with severe conditions given the constraints of the global budget and did not undertake unnecessary treatments. This process is decomposed into two serial sub-processes with an intermediate product, as shown in Figure 1, and thus the efficiencies for the two sub-processes can each be obtained.

### 2.2. Data and Variables

In order to balance the distribution of healthcare resources between urban and rural areas in Taiwan, a National Medical Care Network Program was designed to balance the distribution of healthcare resources and prevent wasting resources in 1985. This changed the administrative boundaries into one medical district in which 17 medical districts are distinguished based on population size, healthcare manpower and facilities, road and transportation systems and the concentration of residents using the same sets of healthcare facilities and resources. Two relatively isolated islands, Kinmen and Matsu, are excluded from the medical care network program, although their residents are insured as part of Taiwan’s NHI program. For this reason, these two islands were used as the 18th medical district in this study.

Although the regional DMUs in this study have no authority governing the allocation of inputs, the hospitals within their regions will adjust the allocation of inputs due to competition and global budgeting (based on the point value). This study is focused on changes in the resource allocation of healthcare services by geographic area, and considering a geographical region based on its related DMUs is a reliable means of measuring this. Both population and healthcare resources are mostly concentrated in DMU B, including Taipei City and New Taipei City (shown on the left of Table 1).

O’Neill, Rauner, Heidenberger and Kraus (2008) provided a cross-national comparison of DEA-based hospital efficiency studies that revealed significant differences with respect to characteristics such as the type of DEA model selected and the choices of input and output categories [24]. In general, input variables refer to the facilities and human resources in a hospital that are used to provide healthcare services, while output variables refer to medical visits, cases, patients, surgeries or inpatient days, which are used to estimate hospital output and thus reflect productivity.

Accordingly, the longitudinal data of the 18 medical districts employed in this work were collected from the annual statistical reports of Taiwan’s Ministry of Health and Welfare (https://dep.mohw.gov.tw/DOS/mp-113.html accessed on 31 July 2019) and used to estimate the efficiency of resource allocation in hospital-based healthcare services under global budgeting, in which the variables used were selected from widely used parameters of healthcare resource efficiency.

With the aim of investigating the longitudinal trend of how global budgeting has influenced the allocation of healthcare resources in Taiwan’s NHI system, it is first necessary to estimate the economic efficiency of hospital-based resource allocation. Accordingly, the efficiency of hospital-based healthcare services is divided into productivity and financial efficiency, which can be independently obtained at different stages to reveal any dynamic changes in efficiency. Figure 3 shows that three input and three output variables in the input-oriented BCC model are included in stage I, and three input and two output variables are in stage II. The detailed definitions of the input and output variables are as follows:
Input Variables
Bed: the total number of hospital beds in a medical district;Physician: the total number of full-time physicians in a medical district;Nursing staff: the total number of full-time nursing staff in a medical district.Output/Input Variables
Outpatient visits: the total number of outpatient visits in a medical district within a year;Emergency room visits: the total number of patients receiving emergency room visits in a medical district within a year;Inpatient days: the total number of patients receiving inpatient treatment services in a medical district within a year.Output Variables
RVU—outpatient: the average RVUs for each outpatient case in a medical district within a year;RVU—inpatient: the average RVUs for each inpatient case in a medical district within a year.


## 3. Results

The main advantages of using DEA for healthcare applications are its flexibility and versatility, as it requires no information on relative prices, and can easily accommodate multiple inputs and outputs. Table 1 summarizes the descriptive statistics of the input and output variables, in which DMUs B, G and M are the top three with the highest amounts of healthcare resources, and DMUs O and R are the bottom two. Without the risk of supply-induced demand, DMUs B, G and M are still the top three areas with the highest output variables of healthcare resource utilization, with DMUs O and R remaining the least. This is mainly due to the geographical disparity in resource allocation, with most healthcare resources concentrated in metropolitan areas.

The BCC model can be used to evaluate the relative efficiency of minimal input consumption for a given level of outputs, or the increase in outputs for given input usage. In addition, basic DEA models provide no insight regarding how the inputs are converted into outputs and whether the operation itself exerts an influence on the overall efficiency. In the two-stage framework used in this study, the efficiency of the whole process can be decomposed into the efficiencies of two sub-processes which are productivity and financial efficiency.

The left part of Table 2 shows the different performance efficiencies of the 18 DMUs from 2003 to 2009. A DMU is relatively efficient only when its efficiency value equals 1. In the first stage, 19 out of the 36 DMUs were found to be relatively efficient, among which 13 of the 18 DMUs were efficient in 2003 after global budgeting was implemented, 6 of which remained efficient in 2009. In particular, the falling efficiencies of DMUs A, F, G, I, M, N and O should be noted, as these were efficient in 2003 and inefficient in 2009. This implies that the productivity of most DMUs fell, with a lower volume of healthcare services being provided after the global budgeting system was introduced.

In a basic DEA model, the financial performance of healthcare services is generally not considered, and we proposed that financial efficiency in the second stage reflect how the hospitals responded to global budgeting by providing appropriate services to patients with severe conditions and not undertaking unnecessary treatments. From the perspective of financial efficiency, only DMU R was efficient in 2003, while DMUs A, B, D, O, P and Q became efficient in 2009, which indicates that the financial efficiency of these DMUs improved from 2003 to 2009.

## 4. Discussion

It is worth noting that Taiwan’s NHI offers almost universal coverage and aims to ensure equal access to healthcare across the different socioeconomic levels of the population. Despite the proposed reform of the reimbursement system to prevent negative outcomes such as the problem of moral hazard, it would be better to focus on improving the efficiency of hospital-based resource allocation rather than cost containment. Countries with successful NHI systems tend to channel all the related resources through a single institution since many multi-payer systems operating without close coordination can lead to waste [25]. With a lack of appropriate cost constraints, health care providers might inflate service demands and carry out unnecessary services, in part because of their asymmetric dominance in terms of medical knowledge [26]. Single-payer insurance systems were able to take advantage of being the sole purchaser to obtain better prices and can exert strict control over expenditure through the use of technology assessment and drug formularies [27].

In order to further explore how the reimbursement system influences healthcare resource allocation in Taiwan’s NHI system, it is essential to estimate the economic efficiency of production and finance. In this study, a two-stage DEA framework was used to decompose the overall efficiency into the efficiencies of two sub-processes—which are productivity and financial efficiency. From the perspective of efficiency evaluation, the decomposition of the productivity of healthcare services into productivity and financial efficiency provides two stages to observe its overall efficiency. The efficiency of a whole process indicates the resource utilization of healthcare services by the given input. In stage I, good productivity means that healthcare inputs are combined to produce a given level of healthcare services with the minimum resources. Financial efficiency in stage II means that the hospitals responded to global budgeting by providing appropriate services to patients with severe conditions and did not undertake unnecessary treatments.

Since efficiency is dynamic, the Malmquist index (MI, see the Appendix A) was selected to measure productivity changes over time, defined as the assimilation of efficiency changes of each unit and technology changes between two periods. To compare the efficiency between 2003 and 2009 in Taiwan’s NHI system, the input-oriented MI is composed of four distance functions to represent multiple inputs and multiple outputs to measure the change in productivity. In addition, MI is used to characterize efficiency since the distance functions are correlated with the technical efficiency measure, and thus this is a natural approach to model a production frontier where the deviations and shifts from the frontier indicate changes in efficiency and technology, respectively.

On the right side of Table 2, MI >1 indicates an improvement in the total factor of productivity in the DMU from 2003 to 2009. Technical efficiency change (catch-up) measures the change in efficiency between 2003 and 2009, while the technological change (innovation) captures the shift in frontier technology. The technique change in this study is the introduction of global budgeting which has improved the performance of the NHI system in Taiwan and further shifted the frontier upwards. The MI of productivity efficiency and the MI of financial efficiency were used to examine the changes in the efficiency of the DMUs from 2003 to 2009. In stage I, DMUs C, D, K, L, P and R improved in efficiency during this period, while in stage II, DMUs B, C, D, I, L, P and Q decreased in efficiency.

From the perspective of efficiency evaluation, DEA is useful for identifying inefficient DMUs that have significant room for improvement. A two-dimensional coordinate is illustrated in Figure 4, which is a combination of the MI and the efficiencies of the two-stage DEA, to explore the impacts of an effective global budgeting system which are more closely related to changes in resource allocation in healthcare services. The space above the *X* axis and along the *Y* axis indicates an improvement in productivity, while the space below indicates a decline. Likewise, the space to the right of the *Y* axis along the *X* axis indicates an increase in financial efficiency, and the space to the left indicates a decrease. The relationship between productivity and financial efficiency is shown in Figure 4, where both quadrants I and III indicate a decrease and increase in productivity and financial efficiency, respectively. DMUs B, I and Q, located in quadrant III, with falls in both productivity and financial efficiency, imply that resource allocation for hospital-based healthcare services worsened during this period. In contrast, DMUs K and R in quadrant I, with increases in both productivity and financial efficiency, show that resource allocation for hospital-based healthcare services improved from 2003 to 2009.

For policymakers, global budgeting leads them to focuses more on productivity than financial efficiency. The DMUs in quadrant I are on target towards better performance in Taiwan’s NHI system. Only DMUs K and R, with higher productivity and financial efficiency, imply that the hospitals located in quadrant I paid attention to rational resource allocation in healthcare services from 2003 to 2009. In this situation, a change in the mix of services offered can lead to better financial efficiency. Similarly, in quadrant II, DMUs C, D, L and P should focus on increasing the RVUs to move towards quadrant I. According to the left part of Table 1, DMUs B, D, G, I, L and M are large metropolitan districts with populations of over 1 million in Taiwan. However, DMUs B, I and Q, which were both inefficient in productivity and financial efficiency, should improve their referral systems and hierarchies of medical services and thus prevent wasting hospitals’ health resources. In addition, DMUs O, P and R, which represent remote areas with populations of less than 250,000, saw improvements in efficiency in productivity and financial efficiency in this study.

Based on the medical care network project, the MOHW restricted the establishment or expansion of hospitals in regions with abundant medical resources to balance medical resource allocation among urban and rural districts. After the introduction of global budgeting, resource allocation in healthcare became more efficient and prevented waste, especially in the metropolitan districts (DMUs B, G and M) with a falling MI with abundant healthcare resources. In contrast, remote districts, such as DMU R, with less hospital-based healthcare resources, demonstrated increased productivity and financial efficiency, with more hospitals being established to close the gap in healthcare between urban and rural districts.

### Limitations

Even though this study has the undeniable merit of offering valuable insights into productivity and financial efficiency, it has some limitations. First, subject to dataset limitations, the improved efficiency in this study cannot be totally attributed to the relationship between hospitals’ behavior and the flexible point-value mechanism in the global budgeting system. Second, due to a lack of clinical data such as those on illness severity, a moral hazard may have occurred in which hospitals changed the incidence of treatment from sicker patients to healthier patients to obtain more RVUs. Third, the two-stage DEA model demonstrated the evolution of the efficiency of the healthcare system during the period 2003–2009, but it is impossible to discuss the impacts of the global budgeting system on the quality of hospital services and on the patients receiving healthcare services.

## 5. Conclusions

In this study, the non-parametric DEA approach was used to assess the effects of the introduction of global budgeting on hospital-based efficiency in Taiwan’s NHI system, with a focus on resource allocation at the macro level, to ensure equitable access to healthcare for the different socioeconomic levels of the population. The results of this study illustrate improvements in the efficiency of hospital resource allocation. Since the NHI system in Taiwan allows all insured individuals complete freedom of choice among hospitals, strong competition can encourage hospitals to enhance both productivity and financial efficiency.

The results of a dynamic analysis using the DEA model demonstrate the effects of global budgeting on resource allocation for hospital-based healthcare services. In addition, the Malmquist index was used with the two-stage DEA model to compare the efficiency during the period from 2003 to 2009, in which one-third of the DMUs improved their efficiency in stage I, and 7 out of the total of 18 DMUs saw falls in inefficiency in stage II. It is notable that the healthcare services in large metropolitan areas with populations of over 1 million, such as DMUs B, D, G, I, L and M, saw only limited improvements after global budgeting, with DMUs D and L only improving in stage I, and DMUs G and M only improving in stage II. Moreover, DMUs B and I saw falls in efficiencies in both stages I and II. In contrast, remote areas with populations of less than 250,000, such as DMUs O, P and R, saw improved efficiency in stages I and II.

In summary, the payment reform in Taiwan’s single-payer system was helpful to reconfigure hospital resource allocation during the period 2003–2009. The two-stage DEA model was easily used to evaluate the efficiency of resource allocation for healthcare policymakers to make the most effective decisions to generate the best outcomes possible.

## Figures and Tables

**Figure 1 ijerph-18-10983-f001:**
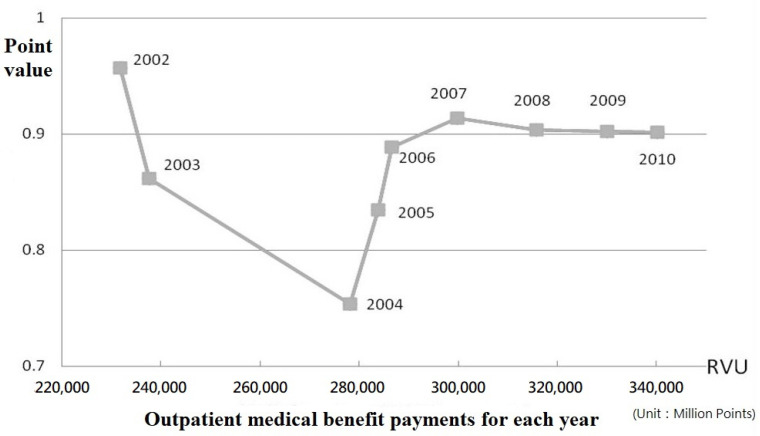
The trend of floating point value in the healthcare sector from 2002 to 2010.

**Figure 2 ijerph-18-10983-f002:**
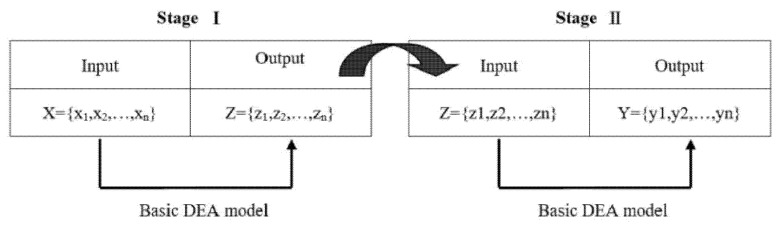
A two-stage DEA model.

**Figure 3 ijerph-18-10983-f003:**
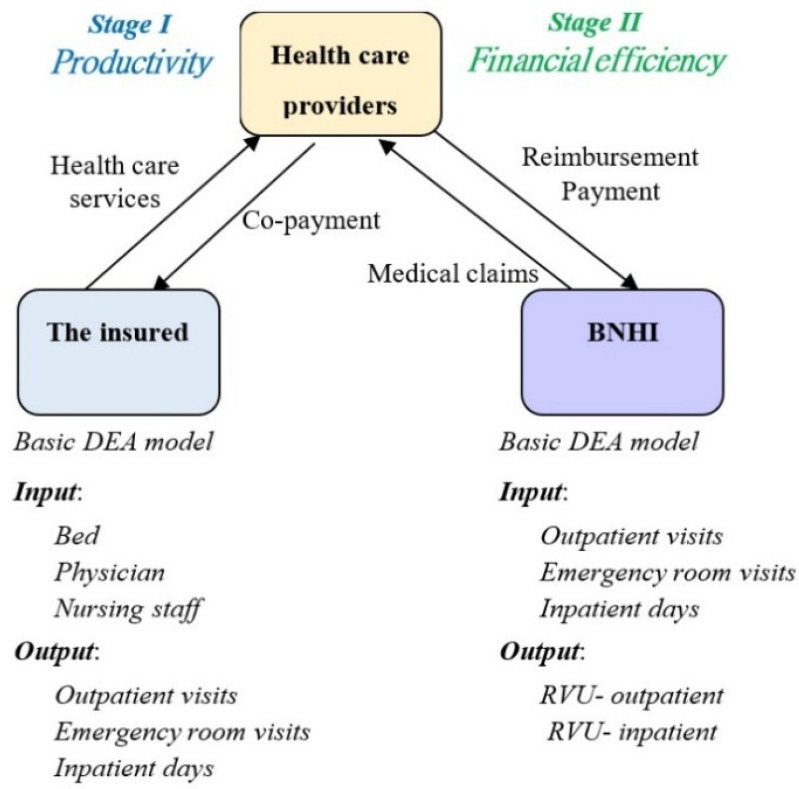
A two-stage DEA model to measure efficiency in Taiwan’s NHI system.

**Figure 4 ijerph-18-10983-f004:**
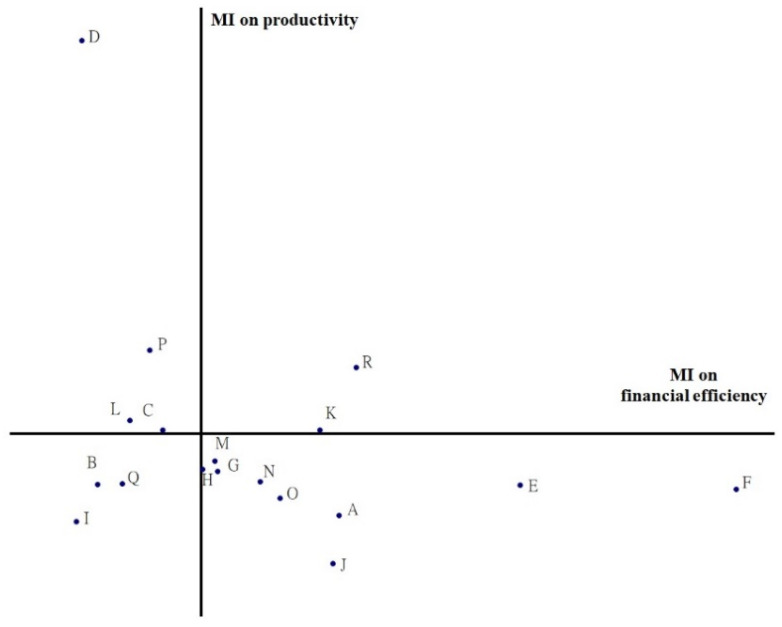
Plot of the Malmquist index (MI) for productivity and financial efficiency.

**Table 1 ijerph-18-10983-t001:** Descriptive information of Taiwan and average input/output variables.

DMU (Medical District)		Descriptive Information of Taiwan						Average Input/Output Variables							
		Area (Unit: km^2^)	Population	(%)	Hospitals	Clinics	Physicians per 1000 People	Bed	Physician	Nursing Staff	Outpatient Visits	Emergency Room Visits	Inpatient Days	RVU—Outpatient	RVU—Inpatient
A.	Keelung district	132.8	388,321	(1.7)	7	270	1.6	2150	353	1154	1,747,912	149,643	409,000	788	43,636
B.	Taipei district	2324.4	6,481,081	(28.0)	99	5814	4.1	31,692	7514	22,781	29,378,496	1,819,952	6,687,438	886	53,533
C.	Yilan district	2143.6	461,625	(2.0)	10	308	1.3	3610	342	1852	1,977,479	173,330	781,214	794	33,354
D.	Taoyuan district	1221.0	1,978,782	(8.6)	35	1363	1.6	11,721	2100	7417	6,704,461	545,123	1,701,739	869	49,522
E.	Hsinchu district	1531.7	922,469	(4.0)	18	708	2.3	4091	520	2442	3,162,070	329,902	632,164	701	36,236
F.	Miaoli district	1820.3	561,744	(2.4)	17	351	0.9	3024	259	1287	2,113,138	132,133	502,797	709	34,517
G.	Taichung district	2214.9	2,635,761	(11.4)	68	3010	4.1	16,291	2790	10,056	12,003,707	764,368	2,893,131	798	45,063
H.	Nantou district	4106.4	530,824	(2.3)	11	407	1.1	2872	250	1310	1,781,903	137,153	615,301	734	28,905
I.	Changhua district	1074.4	1,312,467	(5.7)	36	983	1.4	6154	1040	4115	5,044,491	317,383	1,077,737	773	40,684
J.	Yunlin district	1290.8	722,795	(3.1)	16	492	1.1	2872	334	1427	1,976,836	148,754	425,644	628	33,814
K.	Chiayi district	1963.7	821,577	(3.6)	16	644	3.8	6342	834	3927	4,140,587	294,992	1,078,344	852	46,294
L.	Tainan district	2191.7	1,875,406	(8.1)	37	1696	3.3	9392	1558	5370	6,495,881	494,927	1,657,937	795	47,625
M.	Kaohsiung district	2946.3	2,770,887	(12.0)	94	2585	3.6	16,711	2994	10,301	13,284,762	795,320	3,001,412	819	47,444
N.	Pingtung district	2775.6	882,640	(3.8)	27	623	1.2	5086	539	2917	3,247,647	278,762	977,347	718	34,263
O.	Penghu district	126.9	96,210	(0.4)	3	83	1.3	419	49	210	279,116	41,812	72,709	629	29,468
P.	Taitung district	3515.3	232,497	(1.0)	7	145	1.1	1319	153	709	769,362	80,732	216,612	771	41,554
Q.	Hualien district	4628.6	340,964	(1.5)	11	267	2.2	4170	521	1893	1,453,249	139,279	1,016,509	882	47,728
R.	Kinmen and Matsu	180.5	103,722	(0.4)	2	43	2.2	293	39	150	291,151	27,378	46,288	657	25,615
Min		126.9	96,210	(0.4)	2	43	0.9	272	33	129	258,461	22,318	41,703	538	20,810
Max		4628.6	6,481,081	(28.0)	99	5814	4.1	33,615	8115	25,799	30,400,628	1,969,525	6,798,900	1009	55,205
Mean		2010.5	1,284,431.8	-	25.6	1099.6	2.1	7122.6	1232.5	4406.3	5,325,124.6	370,607.7	1,321,850.9	766.8	39,958.6
SD		1280.4	1,535,232.5	-	29.5	1446.8	1.1	7750.4	1794.4	5525.5	6,963,108.0	424,958.3	1,567,854.7	115.0	9015.3

**Table 2 ijerph-18-10983-t002:** Compare efficiency within two DEA models and the Malmquist index between 2003 and 2009.

DMU (Medical District)		Two-Stage DEA Model and Basic DEA Model						Malmquist Index between 2003 and 2009					
		Two-Stage DEA Model				Basic DEA Model		Stage I (Productivity)			Stage II (Financial Efficiency)		
		Stage I		Stage II		2003	2009	Technical Efficiency Change	Technique Change	Malmquist Index	Technical Efficiency Change	Technique Change	Malmquist Index
		2003	2009	2003	2009								
A.	Keelung district	1.000	0.953	0.505	1.000	1.000	1.000	0.944	0.916	0.865	1.121	1.022	1.145
B.	Taipei district	1.000	1.000	0.232	1.000	1.000	1.000	1.000	0.916	0.916	0.887	1.006	0.892
C.	Yilan district	1.000	1.000	0.247	0.479	1.000	0.396	1.038	0.969	1.006	0.965	0.994	0.960
D.	Taoyuan district	0.575	0.975	0.199	1.000	0.706	1.000	1.694	0.971	1.645	0.871	1.006	0.876
E.	Hsinchu district	1.000	1.000	0.109	0.300	0.200	0.262	0.838	1.092	0.915	1.166	1.144	1.334
F.	Miaoli district	1.000	0.931	0.219	0.609	0.266	0.469	0.976	0.931	0.909	1.551	1.006	1.560
G.	Taichung district	1.000	0.951	0.112	0.256	0.214	0.194	0.992	0.946	0.938	1.012	1.006	1.018
H.	Nantou district	1.000	1.000	0.233	0.484	0.664	0.386	1.000	0.941	0.941	1.106	0.906	1.002
I.	Changhua district	1.000	0.849	0.234	0.244	0.315	0.181	0.868	0.986	0.855	0.864	1.006	0.870
J.	Yunlin district	0.957	0.786	0.225	0.449	0.183	0.336	0.817	0.963	0.786	1.131	1.006	1.138
K.	Chiayi district	0.916	0.844	0.291	0.926	1.000	0.673	1.019	0.987	1.006	1.119	1.006	1.125
L.	Tainan district	0.957	0.931	0.206	0.741	0.394	0.623	1.072	0.953	1.021	0.920	1.006	0.926
M.	Kaohsiung district	1.000	0.970	0.114	0.557	0.229	0.505	1.010	0.945	0.955	1.009	1.006	1.015
N.	Pingtung district	1.000	0.951	0.151	0.224	0.231	0.168	0.916	1.006	0.921	1.048	1.013	1.062
O.	Penghu district	1.000	0.959	0.992	1.000	1.000	0.803	1.000	0.894	0.894	1.000	1.083	1.083
P.	Taitung district	0.768	0.924	0.955	1.000	1.000	1.000	1.144	0.994	1.137	0.877	1.080	0.947
Q.	Hualien district	1.000	1.000	0.799	1.000	1.000	1.000	1.000	0.917	0.917	0.852	1.078	0.918
R.	Kinmen and Matsu	1.000	1.000	1.000	1.000	1.000	1.000	1.000	1.108	1.108	1.000	1.163	1.163

## Data Availability

The datasets used in this study are available from the Ministry of Health and Welfare, Taiwan, on reasonable request.

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
