# Peer review of "The Impact of Global Budgeting on the Efficiency of Healthcare under a Single-Payer System in Taiwan"

_ijerph, 2021, doi:10.3390/ijerph182010983_

Round 1
Reviewer 1 Report
Dear authors,
Thank you for the opportunity of reviewing your paper. The research topic is interesting: the impact of global budgeting on efficiency. The methodology used is also very interesting, clearly described, and properly implemented. Moreover, the description of the Taiwan health system provided on the paper is also very complete and concise.
However, I believe that the paper conclusions are not supported by your results. Additionally, it is not very clear the added-value of the paper relative to existing literature. In my opinion, these two issues are critical and require further clarification. Therefore, I recommend you to review your paper. I propose below a set of major and minor topics that, if properly addressed, might allow surpassing these two issues and improve results robustness.
In my opinion, the major issues that should be addressed are:
- I believe the paper would benefit from clearly highlighting its contribution to the literature. The authors should clearly explain the added value provided by this paper and the main differences relative to remaining studies. When reading the discussion section, the main novelties brought by this paper are not very clear and compelling. I would suggest you to rewrite that and include additional information on what makes your paper special. I suggest you to pick one or two “selling points” and present them clearly. (for instance, I believe that the use of DEA to analyse global budgeting might be one of those points).
- Building on my previous point, I suggest you to rewrite your introduction. In particular, I would like to see clearly what are the goals of your paper, your main hypothesis, and your main contributions. At this point, that discussion is only briefly mentioned in the last paragraph of the introduction…
- I suggest you to carefully revise the wording of the discussion and highlights section. With the existing methodology, you should not provide causal statements. You are finding a longitudinal time trend and its effects on productivity and allocative efficiency. However, there is no reason to attribute such time trend exclusively to the introduction of global budgeting. This brings me to my main point:
- Your methodology is correct to assess efficiency of the health care system. The two stages seem correctly designed. These results show whether units are efficiency or not. If you perform this analysis on an yearly basis, you will be able to analyse whether such efficiency has increased or not. All of these is correct. However, I do not understand why you attribute those changes to the introduction of global budgeting… The introduction of global budgeting might contribute to explain those changes (might contribute, as you point out, to technological change). But, unless I am reading it wrong, it is not necessarily the only factor affecting it! Throughout the paper you make causal statements which, in my opinion, are not backed by your results and research design. If all DMUs have been subject to global budgeting, then you cannot derive the causal impact from your current setting. I suggest you to clearly provide a discussion on the causality of your results. I might have misinterpreted your findings, by I believe that your results show the evolution of the efficiency of the health system – not the impact of global budgeting.
Additionally, some minor could be incorporated to further improve the paper:
- I believe the paper could benefit from a spelling check and some rewriting. I would advise you to write smaller and more concise sentences. Additionally, some typos can be found in the paper. I provide below some non-exhaustive examples:
- In the abstract, the following sentence is redundant: A two-stage Data Envelopment Analysis (DEA) approach is used to measure the efficiency of hospital resource allocation among stakeholders in Taiwan’s NHI system.
- Line 62: was adopt »» was adopted
- Line 69: budge »» budget
- Line 155: recent » recently
- In the abstract, you need to define DMU before using the acronym
- In the introduction, the following sentence needs a reference: “Accordingly, hospitals tended to increase the volume of their services to achieve better revenues, and health care providers may also have offered more extensive and expensive treatments than were needed.” If no Taiwan specific reference is available, you can use applied or theoretical references from key health economics literature
- Could you clarify what is the distinction between general and special beds?
- Is it possible to exceed global budgeting? What happens in that circumstance? How frequent is it?
- In the paper you provide a discussion on the distinction between global budgeting and fee-for-service. It would be interesting to provide some additional discussion on the distinction between global budgeting and other reimbursement schemes (for instance capitation).
- Could you elaborate on the reasons behind your choice to use DEA? In particular, why not using parametric methods, or stochastic frontier analysis? I believe the some discussion on these should be included in the methods section of your paper.
- Can you provide a discussion on the robustness of your results? Are they significantly time-sensitive? How generalizable are your results?
I hope you find these comments useful and that they can contribute to an improved version of your paper. I believe the paper has some potential, as it focuses on a relevant policy question – in an international context of rising health expenditures!
All the best
Author Response
The authors thank the Reviewer for his thoughtful comments and careful review, which helped improve our manuscript.
First, the authors had modified the discussion to present the different productivity and financial efficiency in two-stage DEA model. Second, the authors compared the efficiency changes between 2003 and 2009 period to explain resource allocation under global budgeting reimbursement in Taiwan’s NHI system. According to the design of this study, the authors emphasized two-stage DEA application among 18 medical districts in the discussion section.
Finanlly, flexible point-value mechanism might affect the hospitals’ behavior under global budgeting, there is few direct evidence in this study to support the relationship between hospitals’ behavior and flexible point-value mechanism. Therefore, the limitation was added into the last paragraph of the Conclusion.
Please see the details in the attachment.

Reviewer 2 Report
Review of “The impact of global budgeting on the efficiency of health care under a single payer system in Taiwan”
This paper adopts a two-stage data envelopment analysis (DEA) approach to measure the efficiency of changes in resource allocation after introducing a global budgeting payment scheme. The topic is interesting and meaningful, but the imperfect descriptions of the method and literature make the result unreliable. It could be accepted if the authors solved the following questions.
There are several ways in which the manuscript could be improved:
(1) The page 2, line 71 mentioned “…the budget is then divided and allocated to six health insurance regions…”, which is contradicted with page 4, line 180, that “…17 medical regions are distinguished based on population size…”
(2) On page 12, line 160-161, is there a typo that “…with DMUs G and M improving only in 160 stage II”?
(3) Why did you mention the prospective payment system on page 12, line 171-172? The PPS is different from the flexible point-value mechanism, so that you cannot have a conclusion about PPS.
(4) On page 3, line 133, I recommend that “…with each input and output…” be changed into “…with each output and input…”.
(5) On page 3, your method part, is the equation (1) from which paper? How are equations (1) and (2) related to equation (5) in the appendix part? As functions in the appendix are your primary method, I suggest you incorporate them into the materials and methods part. Also, adding more descriptions about your method part could be helpful.
(6) Why do you only compare efficiency between 2003 and 2009? If you could calculate all years between 2003-2009, you could plot them in one figure to see the evolution trend among the years.
(7) In page 9, line 24, you mentioned nonparametric MI. Have you measured the nonparametric MI? How to measure it?
(8) There are too many typos in your paper, and the literature and related description are not in good order. It would help if you revised them to let the reader better follow your paper and understand its contribution.
(9) The input and output outcomes are inadequate, such as mortality. It is possible that hospitals can change the incidence of treatment from sicker patients to healthier patients to earn more money, but this could be harmful to sicker patients (Dranove et al. 2003).
(10) You only compare the period after establishing global budgeting and flexible point-value mechanism (2003-2009). It could be better to include the pre-period to get a better conclusion that the global budgeting and flexible point-value mechanism do have an effect. If you cannot do that, I will recommend you add a “Limitation” part.
Reference
Dranove, David, Daniel Kessler, Mark McClellan, and Mark Satterthwaite. 2003. “Is More Information Better? The Effects of ‘Report Cards’ on Health Care Providers.” Journal of Political Economy 111 (3): 555–88. https://doi.org/10.1086/374180.
Author Response

(The authors gave the same response as above.)

Round 2
Reviewer 1 Report
Dear authors,
Thank you for the opportunity of reviewing your paper again. I am happy with the answers provided in the cover letter. Thank you for your time and effort!
I was expecting to see a more substantial change in the introduction and discussion section based on my previous comments. In fact, I believe that your contributions are not yet properly highlighted in this new version. Still, this is not my main issue.
In my opinion, a major issue remains to be addressed relative to the previous version. As before, you are attributing the change in productivity over time to the introduction of global budgeting. Though that might be true, it is not necessarily true. In my opinion, your results demonstrate a correlation between increasing productivity and the introduction of global budgeting. However, you claim a causal effect: you argue that global budgeting is leading to higher productivity. Even if that is true, I don’t think your results are enough to prove that. Thus, I maintain my previous comment:
“I do not understand why you attribute those changes to the introduction of global budgeting… The introduction of global budgeting might contribute to explain those changes (might contribute, as you point out, to technological change). But, unless I am reading it wrong, it is not necessarily the only factor affecting it! Throughout the paper you make causal statements which, in my opinion, are not backed by your results and research design. If all DMUs have been subject to global budgeting, then you cannot derive the causal impact from your current setting. I suggest you to clearly provide a discussion on the causality of your results. I might have misinterpreted your findings, by I believe that your results show the evolution of the efficiency of the health system – not the impact of global budgeting.”
With the dataset that you have available, I believe you will not be able to derive this causal relation. But that's not a problem! You can rewrite and present your results in terms of correlation/ association rather than in causal terms. Rewriting your results and providing this discussion on the paper main body would, in my opinion, be enough to overcome this limitation (as long as this limitation is clearly recognized in the paper).
I hope you find these comments useful and that they can contribute to an improved version of your paper.
All the best
Author Response
Thanks to the Reviewer for insightful and helpful comments to improve the paper, the authors would like revise the results as above mentioned.
The authors agree with Reviewer’s view, the causal impacts from our current setting are not supported. As the Reviewer pointed out that there were few direct evidences in this study to support the relationship between hospitals’ behavior and flexible point-value mechanism, even the flexible point-value mechanism might affect the hospitals’ behavior under global budgeting. Therefore, the authors added three research limitations in the last paragraph of the Discussion section. First, subject to dataset limitation, the improved efficiency in this study were not totally attributed to the relationship between hospitals’ behavior and flexible point-value mechanism in global budgeting system. Second, due to lack of clinical data such as illness severity, moral hazard might exist when the hospitals change the incidence of treatment from sicker patients to healthier patients to obtain more RUV. Third, two stage DEA model demonstrates the evolution of the efficiency of the health system during the period of 2003 to 2009, but it is impossible to discuss the impacts of the global budgeting on the quality of hospital services and patients for health care services.
Although the improved efficiency in this study were not attributed to the flexible point-value mechanism in global budgeting system, the authors tried to use two-stage DEA model to evaluate the efficiency changes of resource allocation in Taiwan’s NHI system for health policy makers to make the most effective decisions to generate the best outcomes possible.

Round 3
Reviewer 1 Report
Dear authors,
Thank you for the opportunity of reviewing your paper again. Thank you for your time and effort!
You now identify the causality issue as a main limitation of your paper. This is an important step in the right direction. Still, throughout the paper you make causal statements which, in my opinion, are not backed by your results and research design. The fact that you acknowledge the limitation is very important. However, as mentioned before, I was expecting you to “clearly provide a discussion on the causality of your results” and to “rewrite and present your results in terms of correlation/ association rather than in causal terms”.
Still, as I have pointed out, the specification of this limitation makes this problem transparent to the reader – even though while reading the text some causal statements persist.
All the best